# The Microstructures and Characteristics of NiO Films: Effects of Substrate Temperature

**DOI:** 10.3390/mi13111940

**Published:** 2022-11-10

**Authors:** Guo-Ju Chen, Chih-Ming Lin, Yung-Hui Shih, Sheng-Rui Jian

**Affiliations:** 1Department of Materials Science and Engineering, I-Shou University, Kaohsiung 840, Taiwan; 2Department of Physics, National Tsing Hua University, Hsinchu 30013, Taiwan; 3Department of Applied Physics, National University of Kaohsiung, Kaohsiung 81148, Taiwan; 4Department of Fragrance and Cosmetic Science, College of Pharmacy, Kaohsiung Medical University, 100 Shi-Chuan 1st Road, Kaohsiung 80708, Taiwan

**Keywords:** NiO thin film, XRD, AFM, UV-Vis, contact angle, nanoindentation

## Abstract

The influence of the substrate temperature on the structural, surface morphological, optical and nanomechanical properties of NiO films deposited on glass substrates using radio-frequency magnetron sputtering was examined by X-ray diffraction (XRD), atomic force microscopy (AFM), UV-Visible spectroscopy and nanoindentation, respectively. The results indicate that the substrate temperature exhibits significant influences on both the grain texturing orientation and surface morphology of the films. Namely, the dominant crystallographic orientation of the films switches from (111) to (200) accompanied by progressively roughening of the surface when the substrate temperature is increased from 300 °C to 500 °C. The average transmittance of the NiO films was also found to vary in the range of 60–85% in the visible wavelength region, depending on the substrate temperature and wavelength. In addition, the optical band gap calculated from the Tauc plot showed an increasing trend from 3.18 eV to 3.56 eV with increasing substrate temperature. Both the hardness and Young’s modulus of NiO films were obtained by means of the nanoindentation continuous contact stiffness measurements mode. Moreover, the contact angle between the water droplet and film surface also indicated an intimate correlation between the surface energy, hence the wettability, of the film and substrate temperature.

## 1. Introduction

NiO is a semi-transparent, wide band gap, *p*-type semiconducting material with a cubic sodium chloride (B1) crystal structure [1,2]. The properties of NiO nanocrystals are strongly dependent on the microstructures; therefore, the synthesis of various NiO nanostructures (such as porous nano/microspheres, nanosheets, and nanofibers) has attracted much attention [3,4,5,6,7,8]. In addition to possessing excellent chemical stability, NiO also exhibits intriguing optical, electrical, and magnetic properties. As a result, it has been regarded as a prominent candidate for an array of applications in areas including solar cells [2,9], gas sensors [10], thin film transistors [11], electrochromic devices [12,13,14], and antiferromagnetic spintronics [15,16,17]. However, while most of the research has been concentrated on the abovementioned characteristics of NiO, research on the mechanical properties of these prominent materials have been remaining largely ignored. Since the contact loading during processing or packaging can significantly degrade the characteristics of the material and hence the performance of devices fabricated based on it, thus, in order to fully harvest the promised potential applications, a comprehensive understanding of the mechanical characteristics of NiO films is indispensable, especially when applications involving structural/functional elements used in nano-devices are considered. In this respect, nanoindentation is one of the most versatile tools being ubiquitously used for characterizing the nanomechanical properties of a wide variety of film/substrate and nanostructured systems due to its high efficiency and practice convenience. For instance, the nanomechanical properties (such as the hardness and elastic modulus) [18,19,20,21,22] as well as the elastic/plastic deformation and fracture behaviors [23,24,25,26] of the indented materials can be readily extracted by merely analyzing the load-displacement curves obtained from the indentation measurement.

Another important property relevant to the future practical applications of NiO-based devices is the characteristic of the film surface. Reducing surface energy can hinder the adhesion of airborne contaminants such that they can be effectively removed by the rolling drops due to the hydrophobic characteristic [27], which, in turn, would significantly improve the environmental durability of NiO films. For instance, the hydrophobic surface has been considered one of the critical factors in many applications of optoelectronic devices [28,29]. Consequently, a comprehensive understanding of how the processing parameters correlate to the hydrophobicity or hydrophilicity nature of the resultant surface is also of great importance in designing the functional devices.

To date, several methods for preparing NiO films have been developed, including electron beam evaporation [30], sol-gel method [31], pulsed laser deposition [32,33], and radio-frequency (RF) magnetron sputtering [34,35], etc. Among them, the RF magnetron sputtering [34] has been standing out as the most widely used method for fabricating thin film functional oxides owing to its advantages of high deposition rates, low cost, easy control, and high efficiency for growing films with good quality.

In the present study, the effects of substrate temperature on the structural, surface morphological and optical properties of NiO films deposited on glass substrates by RF magnetron sputtering method are investigated using X-ray diffraction (XRD), atomic force microscopy (AFM), scanning electron microscopy (SEM) and UV-Visible spectroscopy respectively. In addition, the nanomechanical properties of NiO films are measured by nanoindentation using the continuous contact stiffness (CSM) mode. The obtained nanomechanical properties of NiO films are correlated with the crystalline structure, grain size, and surface morphology of the resultant films, which are strongly dependent on the substrate temperature during deposition. Furthermore, the wettability characteristics manifested by the contact angle between the water droplets and film surface indicated that the surface energy of the resultant NiO films is more relevant to the surface roughness than the intrinsic surface energy anisotropy associated with the crystallographic orientations.

## 2. Materials and Methods

The NiO films were prepared by RF magnetron sputtering with a sintered NiO target [36] placed on the sputtering gun (Angstrom Sciences, PA, USA) as the source material. The glass substrates (Corning, Eagle XG, NY, USA) used were kept at various temperatures of 300 °C, 400 °C, and 500 °C. The base pressure of the sputtering chamber was kept at 5 × 10^–6^ torr. During deposition, pure Ar gas was used as the sputtering ion with a working pressure of about 4 mtorr and the input power was 100 W. The deposition time was about 20 min and the thickness of all NiO thin films obtained was about 260 nm.

The crystal structure of the obtained NiO films was analyzed by X-ray diffraction (XRD) using the Panalytical X’Pert diffractometer(Panalytical, Almelo, The Netherlands) with the CuKα radiation (λ = 0.154 nm). The surface morphology and the root-mean-square of the average surface roughness (*R*_rms_) of NiO films were examined using atomic force microscopy (AFM, Topometrix-Accures-II, Topometrix Corporation, Santa Clara, CA, USA). Scanning electron microscopy (SEM, Hitachi S-4700, Hitachi, Tokyo, Japan) was used to analyze the cross-sectional structures of the NiO films. The optical properties were characterized with transmittance measurement by using a Shimadzu UV-2450 UV-Vis spectrophotometer(Shimadzu, Kyoto, Japan). Moreover, the surface wettability of NiO films was monitored using a Ramehart Model 200 contact angle goniometer (ramé-hart instrument, NJ, USA) with deionized water as the testing liquid under ambient conditions.

The nanoindentation measurements were conducted at room temperature using the MTS NanoXP^®^ system (MTS Corporation, Nano Instruments Innovation Center, Oak Ridge, TN, USA). The resolutions of the loading force and displacement are 50 nN and 0.1 nm, respectively. A Berkovich diamond indenter was pressed into NiO films up to an indentation depth of 50 nm. The strain rate varied from 0.01 to 1 s^–1^. An additional harmonic modulation, with the amplitude and frequency being set at 2 nm and 45 Hz, respectively, was simultaneously applied on the indenter to perform the CSM technique [37]. During the measurement process, the indenter was held at the peak load for 10 s before it was completely withdrawn from the specimen to prevent the influence of creep from interpreting unloading characteristics, which were of essential importance in computing the mechanical properties of NiO films. Before each test, it is important to wait until the thermal drift is reduced to below 0.01 nm/s. In order to assure statistical significance, at least 20 indents were conducted on each sample with every indent being separated more than 10 μm apart.

By definition, the hardness is simply described by dividing the applied indentation load by the projected contact area, *H* = *P*/*A_c_*; where *A_c_* is the projected contact area between the indenter and the film surface at a maximum indentation load, *P*. For an ideal Berkovich indenter, the projected area is given by $A_{c}=24.56h_{c}^{2}$ (*h_c_* is the contact depth). The elastic modulus of the sample can be calculated based on the Sneddon expression: $S=2\beta E_{r}\sqrt{A_{c}}/\sqrt{\pi }$ [38]. Here, *S* is the contact stiffness of the material and *β* is a geometric constant, with *β* = 1.00 for the Berkovich indenter, respectively. The reduced elastic modulus, *E_r_*, can be calculated by the following expression:(1)$$E_{r}={\left(\frac{1-v_{f}^{2}}{E_{f}}+\frac{1-v_{i}^{2}}{E_{i}}\right)}^{-1}$$

Here, *v_i_*, *v_f_* and *E_i_*, *E_f_* are the Poisson’s ratio and Young’s modulus of the indenter and the film being measured, respectively. For the diamond indenter tip, *E_i_* = 1141 GPa, *v_i_* = 0.07, and *v_f_* = 0.25 [36] were assumed for all NiO films.

## 3. Results and Discussion

Figure 1 shows the typical XRD patterns of NiO films deposited at various substrate temperatures of 300 °C (pattern (a)), 400 °C (pattern (b)), and 500 °C (pattern (c)), respectively. Several features are immediately noticed in Figure 1. Firstly, all the diffraction peaks appearing in each pattern can be readily indexed with the primary crystallographic orientations of the B1-structured (i.e., NaCl structure) NiO (JCPDS 47-1049) [39]. This is indicative that the films are not only have high crystalline quality but also essentially single-phased. Secondly, although the films appear more or less equiaxial, it is noted that substrate temperature does have noticeable influences on the detailed microstructure of films. For instance, the intensity of the (111) diffraction peak for the substrate temperature of 300 °C (pattern (a)) is relatively larger than that of the (200) peak, while the tendency is reversed for films grown at the substrate temperatures of 400 °C (patterns (b)) and 500 °C (patterns (c)), respectively. The orientation switching might due to the fact that the NiO(200) surface is nonpolar with a surface energy of ~1.74 J/m^2^ compared to ~4.28 J/m^2^ for the polar (111)-terminated surface [40]. Nevertheless, it is noted that this result is in contrast to the tendency reported by Fasaki et al. [32] for NiO films deposited on oxidized Si substrates by pulsed laser deposition, wherein higher substrate temperature appeared to favor (111)-texturing. Moreover, it is also slightly different from the annealing-driven texturing Cu-doped NiO films reported previously [36], wherein the mere effect of increasing the annealing temperature from 300 to 500 °C appeared to be mainly on improving the film crystallinity rather than switching the preferred texturing orientation. Thus, although based on the surface energy argument cited above one might expect the dominance of (200)-texturing in general, it is apparent that the resultant film texturing orientation has been influenced by various relevant parameters in a much more complex manner. Finally, the high crystalline quality as reflected in the sharpness of the diffraction peaks allows us to estimate the average crystalline size and other local microstructural features. We first assume that all NiO films are fully relaxed, then, the average crystalline size (*D*) can be estimated from the full width at half-maximum (FWHM) of a particular diffraction peak by using the Sherrer’s equation [41]; that is, *D* = 0.9 λ/(*β* cos*θ*), where *λ* is the wavelength of X-ray radiation (CuKα, *λ* = 1.5406Å), *θ* is the Bragg angle and *β* is the FWHM of the selected diffraction peak. By using the (200) diffraction peak, the estimated average crystalline sizes of NiO films deposited at the substrate temperatures of 300 °C, 400 °C, and 500 °C are 15, 19, and 24 nm, respectively. Alternatively, by taking into account the strain effect, the Williamson–Hall analysis gives rise to the following expression [36]:(2)$$\beta \cos\theta =\frac{0.9\lambda }{D_{\mathrm{WH}}}+4\varepsilon \sin\theta$$

By using Equation (2) one can plot βcosθ/ λ vs. sinθ/λ to obtain the magnitudes of *D*_WH_ (the intercept) and the local micro-strain *ε* (the slope) for the individual NiO film. Both *D* and *D*_WH_ values obtained for the NiO films investigated in the present study and those reported in the previous report are listed in Table 1 for comparison. It is interesting to note that for films deposited at ambient temperature and then annealed [36] the grain size and associated local micro-strain, *ε*, appeared to be smaller than those in films directly deposited at temperatures same as the annealing temperatures. The fact that the crystalline size of films deposited at 400 °C (~29 nm) is about the same as that obtained by PLD at the same temperature (~32 nm) [32] suggests that the residual kinetic energy of the depositing substances upon landing on the heated substrates have played a more efficient role in facilitating the grain growth.

Figure 2 displays the surface morphology examined by AFM for NiO films deposited at various substrate temperatures, showing a rather dense and homogenous microstructural appearance, albeit an obvious grain size difference can be immediately observed. The surface roughness analysis revealed that the root-mean-square roughness (*R*_rms_) were 2.97 ± 0.2, 5.87 ± 0.4 and 7.35 ± 0.5 nm for NiO films deposited at the substrate temperature of 300 °C, 400 °C, and 500 °C, respectively. It is noted that the values of *R*_rms_ are substantially larger than that of the annealed films [36] listed in Table 1, presumably due to the larger grain sizes described above. The cross-sectional SEM images shown in Figure 2 reveal that the films are all with columnar structures, which might also explain the marked *R*_rms_ increase with a larger effective grain size seen here.

The wettability behavior of the surface is strongly related to the surface morphology of the sample surface [42]. Figure 3 shows the optical images of water droplets on the surface of films deposited at the indicated substrate temperatures. The results are considered one of the direct manifestations of substrate wettability. The contact angles (*θ_CA_*) are measured by depositing a water droplet on the surface of NiO films, and drawing a tangent to the drop at its base. To reduce the measurement error, the data is obtained by averaging ten measurements for each sample. The results clearly indicate that the *θ_CA_* between the water droplet and film surface increases with the film deposition temperature. Since the *θ_CA_* is a prominent parameter widely used in quantifying the surface energy (hence the wettability) [43], it would be interesting to estimate the corresponding surface energy of the films shown in Figure 3. By considering the dispersive force or the van der Waals force across the interface existing between the water droplet and the solid surface and the Fowkes–Girifalco–Good (FGG) theory [44], as $\gamma_{ls}=\gamma_{s}+\gamma_{l}-2\sqrt{\gamma_{s}^{d}\;\gamma_{l}^{d}}$; where $\gamma_{l}^{d}$ and $\gamma_{s}^{d}$ are denoted as the dispersive portions of the surface tension for the liquid and solid surfaces, respectively. Combining Young’s equation [43] with the above FGG expression, employing the nonpolar liquid deionized water (72.8 mJ/m^2^) as the testing liquid, and assuming that $\gamma_{l}^{d}$ is approximately equal to $\gamma_{l}$, the Girifalco–Good–Fowkes–Young equation can be rewritten as: $\gamma_{s}^{d}=\frac{1}{4}\;\gamma_{l}\;\left(\cos\theta_{CA}+1\right)$; with $\gamma_{s}^{d}$ being the surface energy of films. The values of surface energy obtained are 14.7, 13.2 and 10.6 mJ/m^2^ for NiO films deposited at various substrate temperatures of 300 °C, 400 °C and 500 °C, respectively. It is immediately noted that the obtained values of the film surface energy are all about two orders of magnitude smaller than the intrinsic surface energy calculated by Wolf [40], wherein the surface energy for nonpolar NiO(200) and polar NiO(111) surfaces are 1.74 J/m^2^ and 4.28 J/m^2^, respectively. It is apparent that the surface characteristics manifested in the present NiO films must have been dominated by some extrinsic factors. As described above and evidenced in Table 1, higher substrate temperature tends to lead to more equiaxed microstructure, larger grain size, and rougher film surface (i.e., larger *R*_rms_). The question is which of these apparent substrate temperature-related factors is more prominent in determining the observed surface wettability? Bayati et al. [45] suggested that a large amount of air trapped in the gap of nanoislands for rougher films surface might be the primary reason for the increased hydrophobicity observed in films with larger surface roughness. This might be understood by recognizing that the significantly increased contact area at the air/water interface prevented the water droplets from penetrating into the air pockets and, hence, resulting in the larger *θ_CA_*. On the other hand, since the obtained surface energies were all orders of magnitude smaller than the intrinsic ones, the effect of grain size might be more relevant to the resultant surface roughness than the relative areal ratio between the exposed terminated grain orientations.

The optical transmittance spectra of NiO films deposited at various substrate temperatures of 300 °C, 400 °C, and 500 °C are displayed in Figure 4a. As can be seen, the transmittance values of the present NiO thin films vary between 60 to 85% in wavelengths ranging from 500 to 900 nm. Compared with the previous studies by Reddy et al. [35] (40–70%), Al-Ghamdi et al. [46] (50–70%) and Hwang et al. [47] (30–60%), the obtained transmittance for the present RF-sputtered NiO films appeared to have substantially higher transmittance, presumably due to the better crystallinity of the present films. Indeed, extensive efforts had been devoted previously to improving the film transmittance by controlling the amount of gas mixtures during deposition [48], selecting various dopant elements [49], and/or conducting additional annealing treatments [34] on NiO films. As well, all results suggested that, for NiO films prepared by RF-sputtering with Ar as the working gas, appropriate substrate temperature has played a key role in improving the film transmittance. In Figure 4a, it is also noted that the film transmittance exhibits apparent oscillating behavior. In general, the spectral oscillation behavior in thin dielectric films could originate from absorption in combination with interference effects and/or slight thickness non-uniformity in the films [50]. Judging from the cross-sectional SEM images shown in Figure 2, the thickness of the present NiO films appeared to be quite uniform. Thus, the oscillation behaviors in the transmittance spectra are most likely due to the absorption and interference effects. Within the context of the interference scenario, constructive interference will result whenever the condition $t=\left(\frac{2m+1}{2n}\right)\lambda$ is satisfied and, hence, the transmitted radiation will go through a maximum [51]. Wherein *t* is the film thickness, *m* is an integer, *n* is the refractive index of the film, and *λ* is the wavelength of the electromagnetic wave propagating within the film, respectively. Moreover, it is also possible to find *n* by measuring the wavelength at which two adjacent maxima (*λ*_1_ and *λ*_2_) occur through the following expression [51]:(3)$$n=\frac{1}{\left(\frac{1}{\lambda_{2}}-\frac{1}{\lambda_{1}}\right)t}$$

However, if we assume the *n*~2.18 for NiO film [52] and take the transmittance curve (blue) of the 300 °C film as an example, the wavelengths at which the two adjacent maximum transmittances occur are ~430 nm and ~590 nm would give *t* ≈ 730 nm based on Equation (3). At the first glance, this result is nearly 3 times larger than the real film thickness of ~260 nm estimated in Figure 2. However, if we consider that the interference is occurring between the incident radiation and the radiation first reflects at the film/substrate interface and then reflects back again at the film surface, resulting in an effective path difference of about 3 times of actual film thickness, which is quite consistent with the obtained result. Nevertheless, it should be noted that *n* is also a strong function of the wavelength [52] and the effects of absorption were not taken into account in the above simplified zeroth order estimation. This might also explain the seemingly peculiar behavior seen for the 400 °C film. Therefore, although the oscillations in the transmission spectra could be reasonably attributed to the interference effects, quantitative analyses would need more detailed information about the wavelength dependence of the refractive index as well as the absorption in the films.

The other important parameter that can be obtained from the optical spectra is the optical band gap (*E*_g_), which can be calculated from the transmittance data by means of the Tauc equation [53]: $\alpha h\nu =C{\left(h\nu -E_{g}\right)}^{\gamma }$, wherein *a* is the absorption coefficient, *hv* is the photon energy, *C* is the constant, and *γ* is an exponent describing the characteristic of the energy gap, respectively. The exponent *γ* can be 0.5 or 2 depending on whether it is an allowed direct or indirect gap [54]. Figure 4b shows a plot of (*ahv*)^2^ versus photon energy which is linear at the absorption edge, confirming that the NiO films have a direct band gap. The value of band gap is estimated by extrapolation straight line to the linear part to intersect the photon energy axis at *E*_g_ values [53]. At the lower substrate temperature (300 °C), the absorption edge of NiO film ranges from 1.5 eV to 3.5 eV, indicating that the poor crystallinity of films. Moreover, it can be observed that the calculated values of *E*_g_ for the present NiO films suggest that the optical energy gap increases with film crystallinity, which is also in agreement with the previous studies [47,55]. Oh et al. [48] proposed that the band gap of NiO films can be tailored by manipulating the preferred orientation by controlling the amount of nitrogen incorporated into the gas mixtures during film deposition. In this work, a similar trend is observed, namely the transmittance and *E*_g_ values of NiO films are both increased as the preferred (200) orientation becomes more predominant (see Figure 1).

The influences of substrate temperature, hence the film microstructure, on the nanomechanical properties of NiO films are also evaluated by nanoindentation tests. The typical load-displacement curves of NiO films deposited at various substrate temperatures of 300 °C, 400 °C and 500 °C, as displayed in Figure 5. In Figure 5a, the nanoindentation curves provide information about the elastic and plastic deformation behaviors. All the curves appear to be smooth and regular. It is noted that the absence of any discontinuities along either the loading (so-called pop-in event) or unloading (so-called pop-out event) part is in sharp contrast to those observed in Ge thin films [56] and single-crystal Si [57], indicating that neither the formation of crack nor indentation-induced phase transition is involved here. Indeed, no cracking phenomenon is observed in the indented NiO films surface, see Figure 5b.

The total penetration depth into NiO films was approximately 50 nm, which was well within the 30% criterion for avoiding the substrate effects proposed by Li et al. [37]. Thus, the hardness and Young’s modulus of NiO films can be calculated directly from the load-displacement curves (please see Figure 5a) following the analytic method developed by Oliver and Pharr [58]. The values of hardness are 15.7 ± 0.1, 16.6 ± 0.2 and 22.3 ± 0.6 GPa for NiO films deposited at various substrate temperatures of 300 °C, 400 °C and 500 °C, respectively (please see Figure 5c). Moreover, the values of Young’s modulus are 185.4 ± 20.2, 189.6 ± 16.7 and 227.2 ± 21.3 GPa for NiO films deposited at various substrate temperatures of 300 °C, 400 °C and 500 °C, respectively (please see Figure 5d). Compared with the results reported by Fasaki et al. [32], the present results are substantially larger. In particular, the hardness of the present NiO films is nearly twice as large compared to that reported in the previous work [32]. It is noted, however, that, by judging from the intensity and the FWHM of the (200) diffraction peak, the crystallinity of the present NiO films appeared to be much better than that reported in Ref. [32]. Therefore, it is plausible to deduce that the maximum hardness exhibited by the 500 °C NiO film may have intimate correlations with the film crystallinity. Moreover, according to the XRD and nanoindentation results described above, there is a clear tendency showing that films with larger grain size are having larger hardness. The results appeared to exhibit a typical manifestation of the inverse Hall-Petch effect [59]. It has been generally conceived that the Hall-Petch effect in enhancing hardness is primarily governed by hindering the dislocation activities, while the grain boundary sliding is more prominent in accounting for the film hardness displaying the inverse Hall-Petch effect [60,61]. Consequently, the behaviors observed here may be indicative that grain boundary structure is more relevant to the mechanical responses in the present NiO films during nanoindentation.

## 4. Conclusions

In summary, the microstructural, surface morphological, optical, nanomechanical and wetting properties of NiO films deposited on glass substrates at various substrate temperatures were investigated. The XRD results indicated that NiO films had cubic NaCl structure and the films’ crystallinity appeared to be improved with increasing the substrate temperature. Moreover, both the crystalline size and surface roughness of films were also increased with increasing the substrate temperature, which, in turn, lowered the film’s surface energy. In addition, the calculated surface energies were all about two orders of magnitude smaller than the intrinsic surface energy of NiO, indicating that low surface energy and the hydrophobic characteristics were originating mainly from extrinsic factors, such as surface roughness. All NiO films exhibited transmittance ranging from 60% to 85% in the visible wavelength range. The oscillating behavior seen in the transmittance spectra was attributed to the interference effect and the optical energy gap of NiO films increased from 3.18 to 3.56 eV with increasing the substrate temperature, presumably due to the improved films’ crystallinity. From the nanoindentation results, it can be found that the hardness and Young’s modulus of NiO films are increased from 15.7 ± 0.1 to 22.3 ± 0.6 GPa and from 185.4 ± 20.2 to 227.2 ± 21.3 GPa, respectively, as the substrate temperature is increased from 300 °C to 500 °C. Together with the fact of increased film crystalline size, it is indicative that nanomechanical properties might have been more dominated by grain boundary sliding mechanism than by the hindered dislocation activities. Finally, it is concluded that substrate temperature plays a key role in controlling the microstructural, surface morphological, optical, and nanomechanical characterizations of NiO films, as well as their wettability properties.

## Figures and Tables

**Figure 1 micromachines-13-01940-f001:**
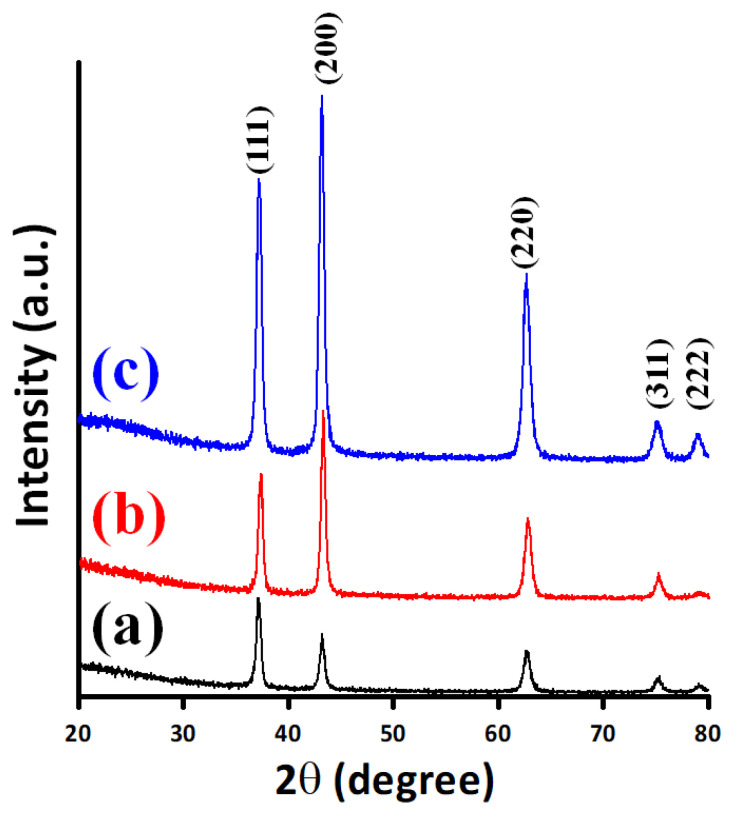
XRD patterns of NiO films deposited at various substrate temperatures of (a) 300 °C, (b) 400 °C, and (c) 500 °C.

**Figure 2 micromachines-13-01940-f002:**
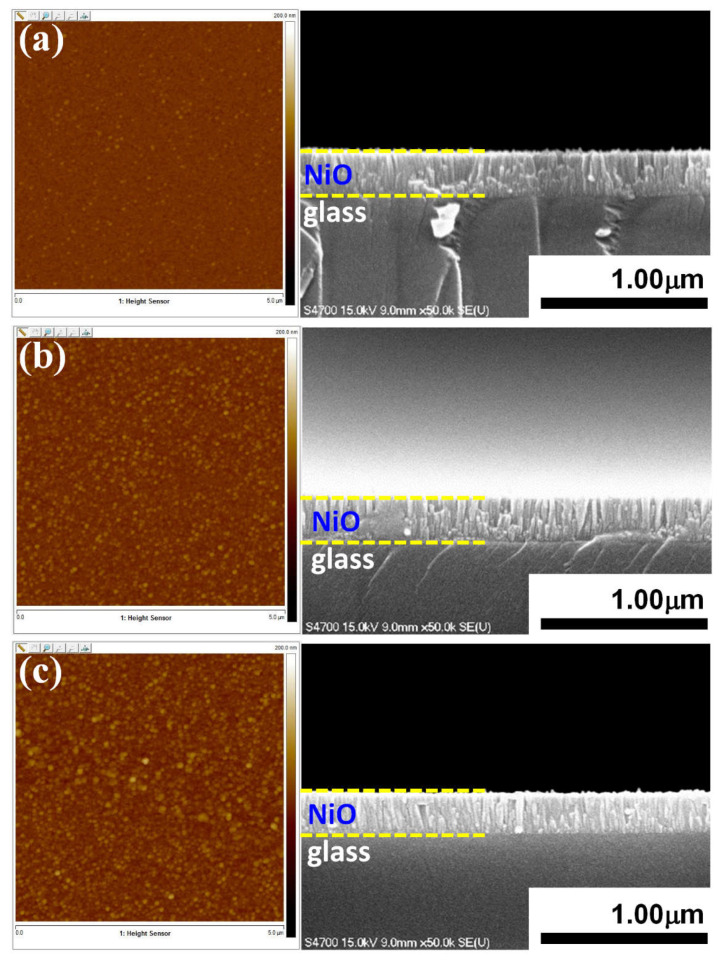
(**left**) AFM surface micrographs (5 × 5 μm^2^) of NiO films deposited at various substrate temperatures: (**a**) 300 °C, (**b**) 400 °C and (**c**) 500 °C. (**right**) The corresponding cross-sectional SEM images.

**Figure 3 micromachines-13-01940-f003:**
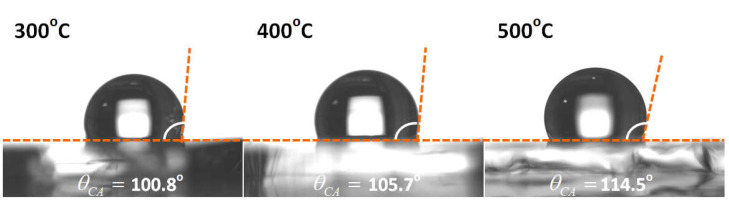
Contact angle images of NiO films surface with water: the substrate temperature of 300 °C, 400 °C, and 500 °C, respectively.

**Figure 4 micromachines-13-01940-f004:**
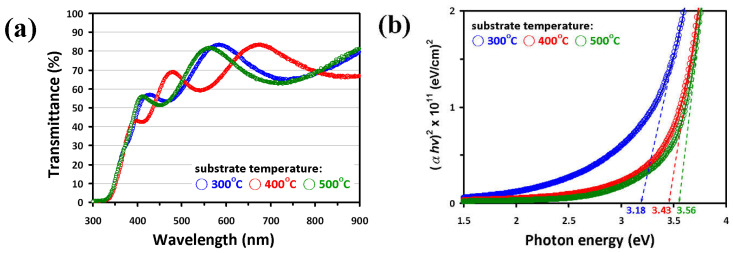
(**a**) UV-Vis transmittance spectra of NiO films with various substrate temperatures. (**b**) Plots of (*αhv*)^2^ versus photon energy of NiO films with substrate temperatures and the evaluated *E*_g_ values of NiO films are shown.

**Figure 5 micromachines-13-01940-f005:**
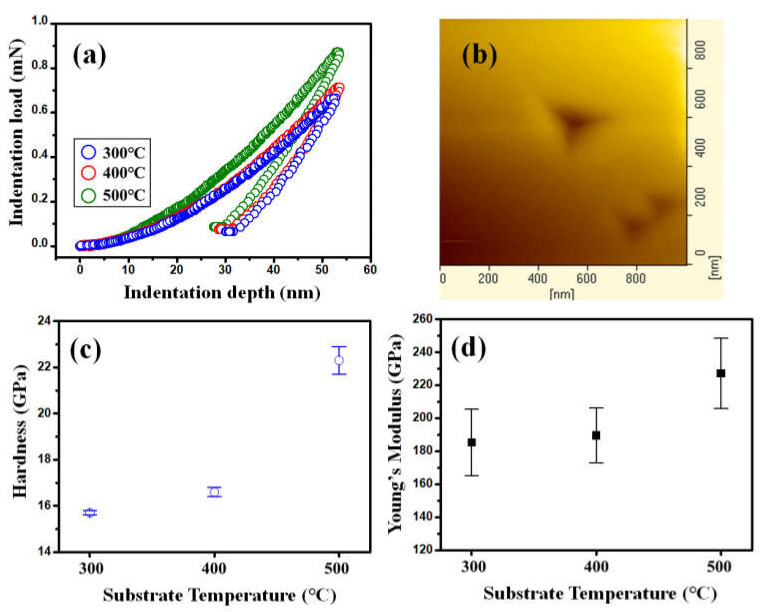
(**a**) Nanoindentation CSM results’ load-displacement curves; (**b**) the obtained indentation image of NiO film at 300 °C-substrate temperature; (**c**) hardness and (**d**) Young’s modulus of NiO films at various substrate temperatures.

**Table 1 micromachines-13-01940-t001:** The structural and surface characteristics of the annealing-driven orientation textured and the present substrate-temperature-induced textured NiO films.

**Annealing driven orientation texturing Cu-doped NiO films [36]**						
Annealing temperature (°C)	Crystalline size (nm) and microstrain *ε* (%)			*R_rms_* (nm)	Contact angle (°)	Surface energy (mJ/m^2^)
	*D*	*D* _WH_	ε			
As-deposited	5.7	10.3	0.65	0.7	45.7	30.9
300	8.4	13.6	0.70	1.4	55.8	28.4
400	11.2	23.8	0.77	2.9	80.4	21.2
500	18.6	38.5	0.80	3.8	97.5	15.8
**NiO films deposited at various substrate temperatures [this work]**						
Substrate temperature (°C)	Crystalline size (nm) and microstrain *ε* (%)			*R_rms_* (nm)	Contact angle (°)	Surface energy (mJ/m^2^)
	*D*	*D* _WH_	ε			
300	5	26	0.70	2.97 ± 0.2	100.8	14.7
400	9	29	0.91	5.87 ± 0.4	105.7	13.2
500	24	41	1.38	7.35 ± 0.5	114.5	10.6

## Data Availability

Not applicable.

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
