# Peer review of "The Microstructures and Characteristics of NiO Films: Effects of Substrate Temperature"

_micromachines, 2022, doi:10.3390/mi13111940_

Round 1

Reviewer 1 Report

This work is important and instructive, but it is necessary to add other elements (doping) in order to improve the different structural, microstructural, optical properties, .... of this material.

Author Response

Dear Reviewer,

Please kindly find the attached file.

Thank you very much!

JIAN

Reviewer 2 Report

The paper investigates different physical properties of NiO thin films, and how they are affected by their temperature. Several things need to be addressed for publication of this article:

1. 260 nm thick films are not really thin films, as they are rather thick.

2. Are the phase changes reversible, when the material is heated?

3. In Figure 5 the load and unload curves have different slopes, indicating structural changes on the surface. This should be addressed in the text.

Author Response

(The authors gave the same response as above.)

Reviewer 3 Report

See attahced.

Author Response

(The authors gave the same response as above.)

Round 2

Reviewer 3 Report

Accept as it is.